# Primary Multi-Systemic Metastases in Osteosarcoma: Presentation, Treatment, and Survival of 83 Patients of the Cooperative Osteosarcoma Study Group

**DOI:** 10.3390/cancers16020275

**Published:** 2024-01-08

**Authors:** Vanessa L. Mettmann, Claudia Blattmann, Godehard Friedel, Semi Harrabi, Thekla von Kalle, Leo Kager, Matthias Kevric, Thomas Kühne, Michaela Nathrath, Benjamin Sorg, Mathias Werner, Stefan S. Bielack, Stefanie Hecker-Nolting

**Affiliations:** 1Cooperative Osteosarcoma Study Group, Paediatrics 5 (Oncology, Haematology, Immunology), Centre for Paediatric, Adolescent and Women’s Medicine, and Stuttgart Cancer Centre, Klinikum Stuttgart–Olgahospital, 70174 Stuttgart, Germany; 2Department of Thoracic Surgery, Faculty of Science, University of Tubingen, 72076 Tubingen, Germany; 3Heidelberg Ion Beam Therapy Centre (HIT), Department of Radiation Oncology, University Hospital Heidelberg, 69120 Heidelberg, Germany; 4Radiologic Institute, Centre for Paediatric, Adolescent and Women’s Medicine, Stuttgart Cancer Centre, Klinikum Stuttgart–Olgahospital, 70174 Stuttgart, Germany; 5St. Anna Children’s Hospital, University Hospital for Paediatric and Adolescent Medicine of the Medical University and St. Anna Children’s Cancer Research Institute (CCRI), 1090 Vienna, Austria; 6University Children’s Hospital Basel, 4031 Basel, Switzerland; 7Department of Paediatrics and Children’s Cancer Research Centre, Klinikum Rechts der Isar, School of Medicine, Technical University of Munich, 81675 Munich, Germany; 8Paediatric Haematology and Oncology, Klinikum Kassel, 34125 Kassel, Germany; 9Osteopathology Reference Centre, Institute of Pathology, Vivantes Klinikum im Friedrichshein, 10249 Berlin, Germany; 10Department for Paediatric Haematology and Oncology, University’s Children’s Hospital Muenster, 48149 Muenster, Germany

**Keywords:** osteosarcoma, multi-systemic metastases, combined metastases, primary metastases, survival

## Abstract

**Simple Summary:**

The prognosis of osteosarcoma patients with primary metastases affecting multiple organ systems is deemed mostly fatal, with survival rates less than 10%. The aim of this study was to identify potential prognostic factors and to evaluate the impact various therapeutic interventions may have on the outcomes of those patients. The poor prognosis was confirmed in our cohort. Surgical resection at all tumour sites was of the utmost importance for long-term survival, while standard chemotherapy was often insufficiently effective. For unresectable bone metastases, radiotherapy might be considered.

**Abstract:**

Background: To evaluate patient and tumour characteristics, treatment, and their impact on survival in patients with multi-systemic metastases at initial diagnosis of high-grade osteosarcoma. Precedure: Eighty-three consecutive patients who presented with multi-systemic metastases at initial diagnosis of high-grade osteosarcoma were retrospectively reviewed. In cases of curative intent, the Cooperative Osteosarcoma Study Group recommended surgical removal of all detectable metastases in addition to complete resection of the primary tumour and chemotherapy. Results: Eighty-three eligible patients (1.8%) were identified among a total of 4605 individuals with high-grade osteosarcoma. Nine (10.8%) of these achieved complete surgical remission, of whom seven later had recurrences. The median follow-up time was 12 (range, 1–165) months for all patients. Actuarial event-free survival after 1, 2, and 5 years was 9.6 ± 3.2%, 1.4 ± 1.4%, and 1.4 ± 1.4%, and overall survival was 54.0 ± 5.6%, 23.2 ± 4.9%, and 8.7 ± 3.3%. In univariate analyses, elevated alkaline phosphatase before chemotherapy, pleural effusion, distant bones as metastatic sites, and more than one bone metastasis were negative prognostic factors. Among treatment-related factors, the microscopically complete resection of the primary tumour, a good response to first-line chemotherapy, the macroscopically complete resection of all affected tumour sites, and local treatment (surgery ± radiotherapy) of all bone metastases were associated with better outcomes. Tumour progression under first-line treatment significantly correlated with shorter survival times. Conclusion: The outlook for patients with multi-systemic primary metastases from osteosarcoma remains very poor. The utmost importance of surgical resection of all tumour sites was confirmed. For unresectable bone metastases, radiotherapy might be considered. In the patient group studied, standard chemotherapy was often insufficiently effective. In the case of such advanced disease, alternative treatment options are urgently required.

## 1. Introduction

Fewer than one out of four to five patients suffering high-grade osteosarcoma have detectable metastases at initial diagnosis [1,2,3,4,5,6]. If there is detectable metastasis at that time, the lungs are by far the most commonly affected site, followed by bones. Other localisations such as lymph nodes, parenchymal organs, or soft tissue are rarely involved [2,3,4,7,8,9]. Treatment for metastatic osteosarcoma usually includes chemotherapy as well as complete surgical resection at all tumour sites (primary and metastatic) [10,11,12,13].

The prognosis of individuals with primary metastasis is significantly worse than that of individuals with localised disease [2,3]. Primary metastases affecting multiple organ systems are deemed mostly fatal. Previous series have found survival rates of less than 10% for affected patients [7,14,15]. Here, we report on a large series of consecutive patients with such multi-systemic metastases at initial presentation of high-grade osteosarcoma. The aim of this study was to identify potential prognostic factors and to evaluate the impact various therapeutic interventions may have on outcome.

## 2. Patients and Methods

### 2.1. Patients

This report includes patients with a primary high-grade osteosarcoma and primary metastases affecting more than one organ system. Patients needed to be registered with the Cooperative Osteosarcoma Study (COSS) between January 1980, and December 2022.

The diagnosis of osteosarcoma needed to have been confirmed histologically. Metastases were required to affect at least two organ systems (lung and bone, lung and other, bone and other, or two distinct organ systems classified as other) and to have occurred within the first four weeks after starting chemotherapy. At least one metastasis of each organ system needed to be strongly suspected radiographically or proven histologically or must have become obvious due to progressive disease. Lesions within the bone of origin of the primary tumour (skip lesions) were not considered metastases.

### 2.2. Initial Diagnostics and Intended Treatment

The primary tumour site was to be investigated by conventional radiography in all studies, whereas the use of computed tomography and magnetic resonance imaging varied over time and depended on availability. Initial staging included an X-ray and/or a computed tomography scan of the chest for detection of possible pulmonary metastases and a ^99^Tc-methylene diphosphonate bone scan and/or positron emission tomography for detection of possible bone metastases. Other organ systems were investigated according to symptoms and local practice. During follow-up, X-rays of the chest and of the primary tumour site were mandatory at regular intervals specified in the respective treatment protocols. The intended first-line treatment included polychemotherapy as well as the surgical removal of all tumour sites, whenever feasible.

### 2.3. Follow-Up and Detection of Recurrence

Regular clinical assessment and X-rays of the former site of the primary tumour as well as the chest were recommended for all patients as part of routine follow-up. Computed tomography of the chest was employed at the treating institutions’ discretion. If a recurrence was suspected, appropriate imaging of the primary tumour site and the chest was recommended, as was a bone scan. The diagnosis of recurrence was based on the assessment of the respective treating institution.

## 3. Ethics Approval and Patient Consent

All studies and registries within COSS were accepted by the appropriate ethics and/or protocol review committees. Upon enrolment in a study or registry within COSS, informed consent was required from all patients and/or their legal guardians, depending on the patients’ age.

## 4. Data Collection and Definition of Variables

Data on patient and tumour characteristics at initial diagnosis and first-line treatment were collected and coded as described previously [2]. The following parameters are in need of further explanation:

Patient and tumour characteristics at initial diagnosis (no later than four weeks after start of chemotherapy): pathological fracture—fracture in the area of the primary tumour or other bone metastases; alkaline phosphatase and lactate dehydrogenase—normal vs. elevated levels before the start of chemotherapy (according to the reference values of the respective laboratory); primary tumour site—localisation at an extremity vs. the trunk vs. the head and neck; number of metastatic sites involved—two (lung and bone, lung and other, bone and other, or other and other) vs. three sites; metastatic sites involved—lung and bone, lung and other, bone and other, or other and other (but affecting two distinct organ systems classified as other sites); laterality of pulmonary metastases—uni- vs. bilateral; pleural effusion—evidence of effusion in the pleural space upon imaging; number of bone and other metastases at initial diagnosis—solitary vs. multiple (two or more), other metastases—any metastases neither classified as pulmonary nor bone metastases; complete surgical resection assessed as possible—according to the assessment of the treating institution and, if available, the assessment of one of a panel of COSS reference surgeons.

Treatment-related factors (regarding the time span between initial diagnosis and, if applicable, recurrence or death, whichever occurred first): resection of the primary tumour—none vs. macroscopically complete resection vs. microscopically complete resection (based on the treating institution’s assessment and, if present, surgery and pathology reports); response to first-line chemotherapy—according to Salzer-Kuntschik et al. [16] (good response = tumour viability below 10%); macroscopically complete resection of metastases—by surgery, and based on the treating institution’s assessment and, if present, surgery and pathology reports; radiotherapy—self-explanatory; therapeutic radioactive medication—injection of a radioactive substance with the aim of achieving a therapeutic effect; tumour progression under first-line treatment—based on the treating institution’s assessment and, if present, on a letter of recommendation by COSS; complete local treatment of all bone metastases—no vs. yes to the statement that all bone metastases were treated with local therapy (radiotherapy and/or surgery); type of local treatment of all bone metastases—none vs. surgery vs. radiotherapy plus, if applicable, surgery (provided that all metastases were treated with local therapy); radiotherapy as part of local treatment of all unresected bone metastases—this was answered with a “yes” if all unresected bone metastases were irradiated.

Follow-up information collected prospectively included the date and site of the first disease recurrence as well as of secondary malignancy, should either have occurred; the date the patient was last known to be alive; and, for deceased patients, the date and cause of death. All relevant information included in this report was reviewed by one of the authors (VLM), and the variables stated in Table 1, Table 2 and Table 3 were coded.

## 5. Statistics

All patients were evaluated retrospectively on an intention-to-treat basis. Median values were given with the range (minimum and maximum), and mean values with the standard deviation. Chi-squared analysis was used to compare unrelated parameters. In survival-time analyses, the date of the diagnostic tumour biopsy was set as the starting point. Event-free survival was calculated until relapse, secondary malignancy, or death, whichever occurred first; overall survival was calculated until death. Patients not achieving surgical remission were assumed to have suffered an event on day 1. Follow-up periods were calculated until the date of last documented information. Survival analyses were performed using the Kaplan-Meier method [17]. The log-rank test was used to compare survival curves [18]. All parameters were investigated by univariate techniques [18]. All *p* values were two-sided, and a *p* value of less than 0.05 was considered significant. Statistical analyses were carried out using SPSS (IBM Corp. Released 2022. IBM SPSS Statistics for Windows, Version 29, NY: IBM Corp., New York, NY, USA).

## 6. Results

### 6.1. Patient and Tumour Characteristics

For detailed information on patient- and tumour-related characteristics, please see Table 1. Eighty-three patients with multi-systemic metastases of high-grade osteosarcoma evident at initial diagnosis were identified. Their median age at diagnosis was 14.8 years (range, 1.7–62.3 years). Forty patients (48.2%) were female. One patient presented with a background of Rothmund–Thomson syndrome.

The disease presented itself at the time of initial diagnosis as follows: Pain was reported by 79/82 (96.3%) patients with appropriate data, and the median interval until diagnostic biopsy was 47 days (range, 3–412 days). Swelling was reported by 60/75 (80.0%) patients with appropriate information, and the median interval was 38 days (range, 1–227 days). A total of 9/79 (11.4%) patients with appropriate data suffered a pathological fracture. Alkaline phosphatase levels were elevated in 56 (82.4%) of 68 cases with known laboratory parameters, and lactate dehydrogenase levels in 50/64 (78.1%). Elevated AP correlated with the presence of bone metastases (*p* = 0.016, chi^2^) but not with their number (*p* = 0.159, chi^2^) or the length of the primary tumour (*p* = 0.720, chi^2^).

The primary tumour was localised in an extremity in 72/83 (86.7%) patients, in the trunk area in ten (12.0%), and in the cranium in one (1.2%). Small tumours (<1/3 of the involved bone) accounted for 23/55 (41.8%) extremity tumours with appropriate information.

A total of 25/82 (30.5%) patients presented with metastases in three organ systems, while only two sites were involved in 57/82 (69.5%). One further patient had both pulmonary and other metastases, but there was no information about whether bone metastases were also present. Pulmonary metastasis occurred in 81/83 (97.6%) patients (30 histopathologically verified), and pleural effusions in 9/44 (20.5%) patients with relevant data. A total of 69/82 (84.1%) patients with appropriate information presented with bone metastases (20 histopathologically verified), and 41/83 (49.4%) patients presented with metastases outside the lungs or bone (18 histopathologically verified). The latter involved distant lymph nodes in twenty-three (56.1%) patients, soft tissue in nineteen (46.3%) patients, the heart or pericardium in four (9.8%) patients, and the brain in one (2.4%) patient (multiple mentions possible).

At initial diagnosis, complete surgical resection at all tumour sites was deemed possible in 16/61 (26.2%) patients with appropriate data.

### 6.2. Treatment Strategy at Initial Disease Presentation

Details on treatment strategy are summarized in Table 2.

Macroscopically complete resection of the primary tumour was performed in 35/78 (44.9%) patients with appropriate data, with a good response to neoadjuvant chemotherapy in 11/26 (42.3%) evaluable cases. Macroscopically complete resection of all metastases was achieved in 9/81 (11.1%) patients with appropriate information. A total of 9/83 (10.8%) patients achieved a complete resection of all tumour sites.

Radiotherapy was performed in 27/73 (37.0%) patients with appropriate information, and therapeutic radioactive medication was administered to 9/79 (12.5%) patients with known data.

Local therapy for bone metastases was analysed in more detail and is summarised in Table 3. A total of 12/58 patients received local treatment of all detectable bone metastases. It was performed by surgery in only 9/66 (13.6%) patients with appropriate information and by (additional) radiotherapy in 3/58 (5.2%) patients with available data.

All patients of our cohort received chemotherapy. Twenty-six/83 (31.3%) did so with a delay of more than three weeks. Osteosarcoma was progressive under first-line treatment in 48/66 (72.7%) patients with appropriate information. Tumour progression correlated with response to chemotherapy (*p* = 0.002, chi^2^): of 23 patients with appropriate data, progressive disease was observed in 3/10 (30.0%) good responders and in 12/13 (93.2%) poor responders. Information on progressive disease was available for 44/58 (75.9%) patients with unknown response to chemotherapy: Tumour progression was reported in 34 (77.3%) of these patients. There was no correlation between a delay of treatment and progressive disease (*p* = 0.159, chi^2^). Second-line chemotherapy was known to have been given in 50/83 (60.2%) patients, of whom eleven had two changes of the systemic therapy regime; six had three; and one each had four, five, and six.

### 6.3. Prognostic Factors

Regarding factors present at diagnosis, elevated alkaline phosphatase (*p* = 0.019) correlated with poorer survival. There was a trend towards better survival when lactate dehydrogenase levels were normal; however, this was not statistically significant (*p* = 0.078). Survival was worse in patients with bone and pulmonary metastases (*p* = 0.005, Figure 1A). There was no difference in survival according to the number of metastatic organ systems involved (two vs. three, *p* = 0.839). Pleural effusion (*p* = 0.010) and the appearance (*p* = 0.010) and number of bone metastases (solitary vs. multiple, *p* = 0.028) as well as the absence of “other” metastases correlated with poorer survival (*p* = 0.018). Those patients for whom complete resection at all tumour sites was assessed as possible at initial diagnosis fared significantly better (*p* < 0.001).

Treatment-related factors associated with better outcomes were microscopically complete resection of the primary tumour (*p* < 0.001); a good response to first-line chemotherapy (*p* = 0.044); (macroscopically) complete resection of all metastases (*p* = 0.001, Figure 1B), as well as of all pulmonary and other metastases on an individual basis (*p* < 0.001); and complete local therapy (surgical resection and/or radiotherapy) of all bone metastases (*p* = 0.010). Overall survival according to type of local treatment of all bone metastases is shown in Figure 1C). There was no difference in survival for patients receiving radiotherapy as part of local treatment for all unresected bone metastases (*p* = 0.250); there was a trend toward better survival when all bone metastases were resected, but this was not statistically significant (*p* = 0.062). Patients receiving therapeutic radioactive medication fared worse (*p* = 0.011). Tumour progression under first-line treatment correlated with poorer survival (*p* < 0.001).

In summary, elevated alkaline phosphatase before chemotherapy, pleural effusion, distant bones as metastatic sites, and the presence of more than one bone metastasis were negative prognostic factors. Among treatment-related factors, microscopically complete resection of the primary tumour, a good response to first-line chemotherapy, and macroscopically complete resection of all affected tumour sites as well as local treatment (surgery ± radiotherapy) of all bone metastases were associated with better outcomes. Tumour progression under first-line treatment significantly correlated with shorter survival times.

### 6.4. Survival and Follow-Up

The median follow-up was 12 (range, 1–164) months for all patients and 18 (range, 1–164) months for patients still alive at last contact. The event-free survival rates for all patients at one, two, and five years after initial diagnosis were 9.6 ± 3.2%, 1.4 ± 1.4%, and 1.4 ± 1.4%, and the corresponding overall survival rates were 54.0 ± 5.6%, 23.2 ± 4.9%, and 8.7 ± 3.3% (Figure 2).

At last follow-up, nine (10.8%) patients out of the total sample of eighty-three had achieved a first surgical remission as defined. Two of these remained in their first remission (1.1 and 12.7 years after diagnosis), four died following their first recurrence, two died following their second recurrence, and one was in a third complete remission (6.2 years after diagnosis). Of seventy-four (89.2%) patients never achieving a first complete remission, sixty-three died of osteosarcoma, one due to unknown causes, and one due to septic shock in neutropenia. Of twelve survivors, the follow-up was longer than five years in five patients: In two of those patients (alive for 6.9 and 8.5 years after diagnosis), all remaining bone metastases had been irradiated. One patient (alive at 13.7 years) was under continuous therapy with interferon-alpha. One survivor was in a first remission at 12.7 years and one in a third remission at 6.2 years after initial disease.

## 7. Discussion

This study, the first to report on such a large patient cohort, confirms the exceedingly poor prognosis of patients in whom several organ systems are affected by primary osteosarcoma metastases [7,14,15]. Only one in ten patients in our cohort had achieved a first surgical remission, and one in fourteen patients survived for more than five years after osteosarcoma diagnosis.

A poor prognosis for patients with extra-pulmonary osteosarcoma metastases, including those to the bones and other sites, has already been reported, both by our study group and by others [7,8,9,14,15,19]. Based on the patients examined in this study, it can further be deducted that patients with bony metastases in addition to pulmonary metastases fare worse than those with lesions at other sites in addition to pulmonary lesions.

A comparatively high number of our patients presented with elevated AP values at initial diagnosis. This was to be expected, as it has been reported that elevated AP values correlate with the occurrence of metastases [20]. The predictive value of elevated AP levels seems to be maintained even within the group of patients with metastatic disease—possibly as an indicator of a higher tumour burden [14,20,21]. Bone metastases in particular are reported to be associated with elevated AP values [22]. This observation was confirmed in our cohort. Ultimately, one could conclude that an increase in AP values could possibly be a consequence of the presence of prognostically poor bone metastases.

It was not surprising that the presence of pleural effusion was accompanied by a fatal prognosis: in the presence of malignant pleural effusion, it can be assumed that tumour cells have already spread within the pleural space. Complete surgical remission is then hardly conceivable or even impossible. In concordance, Saoud et al. reported a poor prognosis for patients with malignant pleural effusion in the presence of sarcoma. Both of their patients with pleural effusion accompanying osteosarcoma survived only a few months [23].

Regarding bony involvement, solitary metastases were found to be associated with a better prognosis than more extensive disease. As multiple metastases pose a greater challenge in terms of local treatment, this is not unexpected. A higher number of pulmonary metastases or metastases in general has repeatedly been associated with poorer survival [3,7,15,24,25,26]. However, we could not find a survival benefit for patients with either solitary metastasis to other sites or unilateral lung involvement. For the former, the possibility of a lower detection rate of “other” metastases (due to the lack of standard diagnostics in this regard) may have led to an incorrect classification. For the latter, the small number of cases of unilateral pulmonary metastases might be an explanation.

The initial assessment regarding the possibility of achieving complete remission proved to be significant in terms of survival. However, on closer inspection, it can be deduced that this correlated with short-term survival only. Five-year outcomes, in contrast, were almost the same regardless of whether remission was considered feasible. The reason for this might not be so much that the initial assessment of resectability was incorrect, but rather that some individual patients became long-term survivors despite incomplete surgical resections.

Nevertheless, the high significance of complete surgical resection for further survival was confirmed in our cohort—both resection of the primary tumour and resection of all metastases [2,7,8,10,11,12,13,14,24,27].

In the case of pulmonary metastases, an open thoracotomy is usually advisable, as not all lung metastases can be reliably detected by imaging [28,29,30,31,32]. In addition to the removal of all conspicuous foci, lesions that elude imaging can then be detected by manual exploration [33]. In patients with multiple metastases, however, complete resection of all metastatic lesions is naturally often difficult if not impossible. Alternative therapeutic approaches are particularly important for these patients.

In the particular case of bony metastases, only a trend towards a better prognosis with complete surgical resection of these metastases could be demonstrated. This may be due to the rather limited cohort size—we can only surmise that statistical significance could be achieved with a higher number of cases. There was no advantage at all for patients who received radiotherapy for all non-resected bone metastases. It should be noted here that such a constellation was extremely rare in our cohort (*n* = 3)—it is possible that a higher number of cases could also reveal a significant advantage for patients receiving such therapy. However, if any local treatment was performed for all bone metastases, regardless of whether it was a surgical or radiotherapeutic procedure, the survival rate was significantly higher than for metastases that were not or only partially treated. Furthermore, two of our patients with some bone metastases only treated by radiotherapy were among the few long-term survivors (6.9 and 8.5 years). This also suggests a possible benefit of adequate radiotherapy. Various studies suggest that radiotherapy with appropriate doses, achieved particularly by heavy ions and/or protons, might achieve long-term local osteosarcoma control [34,35,36,37,38,39]. This report indicates that radiotherapy might be a therapeutic alternative for those bony metastases that are not suitable for surgical removal.

It must, however, be noted that no benefit of radiotherapy in general could be detected in our total cohort. The reason for this observation is probably the predominant use of this treatment modality in clearly palliative situations. Therefore, a selection bias must be assumed. The administration of radioactive medications was even associated with worse survival. It seems that patients receiving such therapy were highly selected in the same manner as described above. Altogether, long-term survival was not observed in any patient who received internal radiotherapy. This finding is in line with past reports on internal radiotherapy [40,41].

All of our patients received chemotherapy. As expected, tumour progression under first-line therapy led to a very poor prognosis, with no survivors beyond two years after initial diagnosis. It must be noted that progressive disease did not necessarily occur during actual chemotherapy itself but might also have been caused by delays in the onset of treatment or by prolonged chemotherapy interruptions. However, no clear correlation between chemotherapy delay and tumour progression could be established.

Histological response to upfront chemotherapy is a well-known predictive factor for reduced overall survival [2,7,8]. In our cohort, it was poor in nearly 60% of the evaluable cases. Compared to a poor response in just over 40% of patients with osteosarcoma in general, this seems rather high [2]. In those patients with response data and progressive disease, we even observed an almost uniformly poor tumour response. Since, for patients without information on tumour response, tumour progression under therapy was reported in the majority of cases, in total, an even higher proportion of poor responders than indicated can be assumed. All in all, the overwhelmingly high rate of poor response impressively illustrates that administered systemic therapy appeared to be of very limitedly value in the examined subgroup of patients. Ultimately, the question arises as to whether chemotherapy is justified in this subgroup, as it shows little effect overall. Still, in some patients, disease stabilization or even a good response to systemic treatment was observed. Therefore, in the absence of superior alternatives, first-line chemotherapy should not be abandoned, at least initially. If the tumour is progressive and there are no curative treatment options, a palliative approach might then also be discussed instead of intensive chemotherapy—for example, a therapy attempt with a tyrosine kinase inhibitor might be considered. In phase II trials, cabozantinib, regorafenib, and sorafenib were able to extend the progression-free interval in treatment-refractory or relapsed osteosarcoma patients [42,43,44,45,46,47].

Over the last three decades, no significant improvement in survival has been achieved in primary or metastasised osteosarcoma [12,47,48,49]. The survival rate of patients examined in various clinical trials evaluating new therapies such as immunotherapy, tyrosine kinase inhibitors, and other drug therapies has increased only slightly at best [47,49,50,51]. However, for patients with multisystemic osteosarcoma, new therapeutic approaches are urgently needed, as standard therapy is clearly not sufficient.

## 8. Conclusions

In conclusion, this report confirms the extremely poor prognosis for patients with multi-systemic primary metastases of osteosarcoma. Surgical resection of all tumour sites is of the utmost importance for long-term survival. For unresectable bone metastases, radiotherapy might be considered. In the patient group studied, standard chemotherapy was often insufficiently effective. More effective treatment options for affected patients are direly needed.

## Figures and Tables

**Figure 1 cancers-16-00275-f001:**
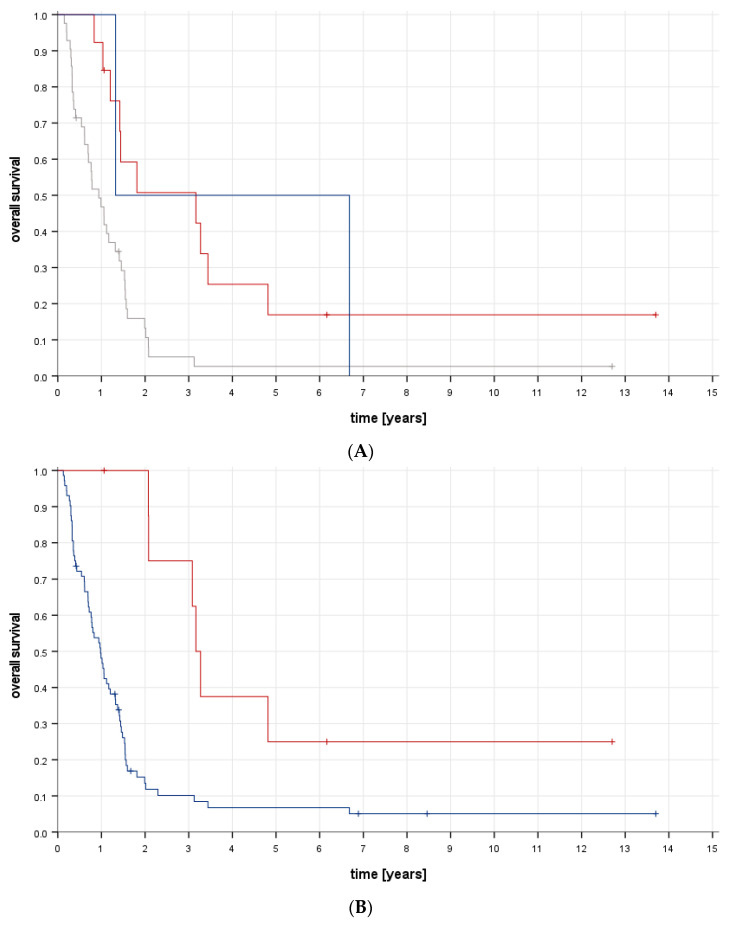
(**A**). Overall survival according to metastatic sites involved at initial diagnosis: lung and other (red; *n* = 13), lung and bone (grey; *n* = 42), bone and other (blue; *n* = 2); *p* = 0.005; log-rank test. (**B**). Overall survival according to achievement of complete resection of all metastases: macroscopically complete (red; *n* = 9), macroscopically incomplete (blue; *n* = 72); *p* = 0.001; log-rank test. (**C**). Patients with bone metastases as part of dissemination: overall survival according to type of local treatment of all bone metastases; surgery only (red; *n* = 9), radiotherapy ± surgery (grey; *n* = 3), no remission (blue; *n* = 46); *p* = 0.037; log-rank test.

**Figure 2 cancers-16-00275-f002:**
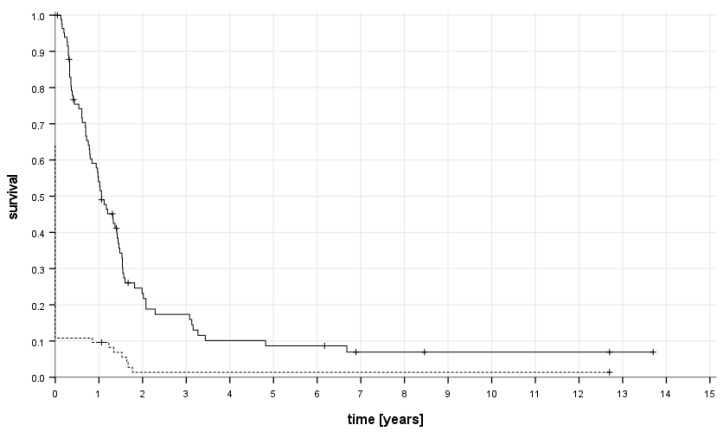
Overall (solid line) and event-free survival (dashed line) of all 83 patients.

**Table 1 cancers-16-00275-t001:** Survival estimates by patient- and tumour-related factors of 83 registered osteosarcoma patients with primary multi-systemic metastases.

				Overall Survival						
	Patients		%	1-Year		2-Year		5-Year		
				Rate	SE	Rate	SE	Rate	SE	*p* *
All eligible patients		83	100	0.540	0.056	0.232	0.049	0.087	0.033	
Age at initial diagnosis, years										
	<14.8	41	49	0.455	0.079	0.278	0.071	0.111	0.052	0.960
	≥14.8	42	51	0.627	0.076	0.177	0.064	0.059	0.040	
Sex										
	Female	40	48	0.479	0.081	0.160	0.060	0.096	0.050	0.311
	Male	43	52	0.597	0.076	0.304	0.074	0.083	0.046	
Duration of pain until initial diagnosis										
	<47 days	35	50	0.460	0.087	0.246	0.075	0.140	0.063	0.773
	≥47 days	35	50	0.591	0.084	0.217	0.076	0.072	0.049	
	No pain/unknown	13								
Duration of swelling until initial diagnosis										
	<38 days	27	50	0.622	0.095	0.311	0.091	0.133	0.070	0.559
	≥38 days	27	50	0.542	0.098	0.152	0.079	0.051	0.049	
	No swelling/unknown	29								
Pathological fracture at initial diagnosis										
	No	70	89	0.537	0.060	0.248	0.053	0.099	0.038	0.402
	Yes	9	11	0.292	0.173	0.146	0.135	0.000	0.000	
	Unknown	4								
Alkaline phosphatase before start of chemotherapy										
	Normal	12	18	0.750	0.125	0.536	0.156	0.214	0.133	0.019
	Elevated	56	82	0.456	0.068	0.152	0.049	0.057	0.032	
	Unknown	15								
Lactate dehydrogenase before start of chemotherapy										
	Normal	14	22	0.929	0.069	0.314	0.129	0.079	0.075	0.078
	Elevated	50	78	0.388	0.071	0.186	0.058	0.093	0.044	
	Unknown	19								
Primary tumour site										
	Extremity	72	88	0.503	0.060	0.241	0.053	0.069	0.033	0.525
	Trunk	10	12	0.778	0.139	0.111	0.105	0.111	0.105	
	Head and neck	1		1.000		1.000		1.000		
Tumour size at initial diagnosis (limb only)										
	Small (<1/3 of the involved bone’s length)	23	42	0.595	0.104	0.250	0.095	0.062	0.059	0.364
	Large (≥1/3 of the involved bone’s length)	32	58	0.488	0.090	0.199	0.076	0.079	0.053	
	Other site/unknown	28								
Number of metastatic sites involved at initial diagnosis										
	Two	57	70	0.529	0.066	0.232	0.058	0.077	0.037	0.839
	Three	25	30	0.438	0.103	0.246	0.093	0.123	0.077	
	Unknown	1								
Metastatic sites involved at initial diagnosis										
	Lung and bone	42	74	0.468	0.078	0.133	0.055	0.027	0.026	0.005
	Lung and other	13	23	0.923	0.074	0.508	0.144	0.169	0.109	
	Bone and other	2	4	1.000		0.500	0.354	0.500	0.354	
	Other and other	0	0							
	Three sites/unknown	26								
Pulmonary metastases at initial diagnosis										
	No	2	2	1.000		0.500	0.354	0.500	0.354	0.428
	Yes	81	98	0.529	0.056	0.225	0.049	0.075	0.032	
Laterality of pulmonary metastases										
	Unilateral	4	5	0.750	0.217	0.500	0.250	0.000	0.000	0.637
	Bilateral	71	95	0.519	0.061	0.215	0.052	0.083	0.035	
	None/unknown	8								
Pleural effusion at initial diagnosis										
	No	35	80	0.441	0.085	0.173	0.069	0.069	0.047	0.010
	Yes	9	20	0.222	0.139	0.000	0.000	0.000	0.000	
	Unknown	39								
Bone metastases at initial diagnosis										
	No	13	16	0.923	0.074	0.508	0.144	0.169	0.109	0.010
	Yes	69	84	0.473	0.061	0.182	0.049	0.073	0.034	
	Unknown	1								
Number of bone metastases at initial diagnosis										
	One	14	21	0.701	0.126	0.390	0.136	0.156	0.101	0.028
	At least two	53	79	0.393	0.068	0.133	0.051	0.053	0.035	
	None/unknown	16								
Other metastases at initial diagnosis										
	No	42	51	0.468	0.078	0.133	0.055	0.027	0.026	0.018
	Yes	41	49	0.617	0.078	0.338	0.078	0.154	0.062	
Number of other metastases at initial diagnosis										
	One	17	43	0.765	0.103	0.499	0.128	0.166	0.105	0.476
	At least two	23	58	0.525	0.109	0.239	0.093	0.143	0.077	
	None/unknown	43								
Complete surgical resection assessed as feasible at initial diagnosis										
	No	45	74	0.387	0.073	0.137	0.052	0.091	0.043	0.010
	Yes	16	26	0.938	0.061	0.554	0.138	0.092	0.087	
	Unknown	22								

* *p*-value, log-rank test.

**Table 2 cancers-16-00275-t002:** Survival estimates by treatment-related factors of 83 registered osteosarcoma patients with primary multi-systemic metastases.

				Overall Survival						
	Patients		%	1-Year		2-Year		5-Year		
				Rate	SE	Rate	SE	Rate	SE	*p* *
Resection of the primary tumour										
	None or macroscopically incomplete	43	55	0.326	0.071	0.051	0.035	0.026	0.025	<0.001
	Macroscopically complete, microscopically incomplete	3	4	1.000		0.000	0.000	0.000	0.000	
	Microscopically complete	32	41	0.775	0.075	0.495	0.093	0.152	0.069	
	Unknown	5								
Response to first-line chemotherapy †										
	Good (grades 1–3)	11	42	1.000		0.727	0.134	0.364	0.145	0.044
	Poor (grades 4–6)	15	58	0.786	0.110	0.397	0.136	0.000	0.000	
	Unknown/primary or no tumour resection	57								
Macroscopically complete resection of all metastases										
	No	72	89	0.481	0.059	0.135	0.042	0.068	0.032	0.001
	Yes	9	11	1.000		1.000		0.250	0.153	
	Unknown	2								
Macroscopically complete resection of all pulmonary metastases										
	No	68	87	0.450	0.061	0.111	0.041	0.056	0.030	<0.001
	Yes	10	13	1.000		1.000		0.222	0.139	
	No pulmonary metastases/unknown	5								
Macroscopically complete resection of all bone metastases										
	No	57	86	0.424	0.065	0.129	0.046	0.064	0.035	0.062
	Yes	9	14	0.778	0.139	0.519	0.176	0.130	0.121	
	No bone metastases/unknown	17								
Macroscopically complete resection of all other metastases										
	No	25	68	0.400	0.098	0.040	0.039	0.000	0.000	<0.001
	Yes	12	32	1.000		0.900	0.095	0.450	0.166	
	No other metastases/unknown	36								
Radiotherapy										
	No	46	63	0.478	0.074	0.205	0.061	0.046	0.031	0.352
	Yes	27	37	0.593	0.095	0.244	0.085	0.163	0.074	
	Unknown	10								
Therapeutic radioactive medication										
	No	63	88	0.551	0.063	0.273	0.058	0.109	0.042	0.011
	Yes	9	13	0.222	0.139	0.000	0.000	0.000	0.000	
	Unknown	11								
Duration until start of chemotherapy, days										
	<21	57	69	0.500	0.067	0.206	0.055	0.056	0.031	0.289
	≥21	26	31	0.636	0.097	0.300	0.099	0.180	0.089	
Tumour progression under first-line treatment										
	No	18	27	0.941	0.057	0.627	0.121	0.314	0.116	<0.001
	Yes	48	73	0.396	0.071	0.090	0.043	0.000	0.000	
	Unknown	17								

* *p*-value, log-rank test. † According to Salzer-Kuntschik et al. [16].

**Table 3 cancers-16-00275-t003:** Survival estimates by treatment-related factors of 69 registered osteosarcoma patients with primary multi-systemic metastases including bone lesions.

			Overall Survival						
	Patients		1-Year		2-Year		5-Year		
			Rate	SE	Rate	SE	Rate	SE	*p* *
All eligible patients with bone metastases		69	0.473	0.061	0.182	0.049	0.073	0.034	
Complete local treatment of all bone metastases									
	No	46	0.370	0.071	0.095	0.045	0.024	0.023	0.010
	Yes (radiotherapy and/or surgery)	12	0.667	0.136	0.476	0.150	0.190	0.120	
	Unknown	11							
Type of local treatment of all bone metastases									
	No	46	0.370	0.071	0.095	0.045	0.024	0.023	0.037
	Surgery only	9	0.778	0.139	0.519	0.176	0.130	0.121	
	Radiotherapy involved	3	0.333	0.272	0.333	0.272	0.333	0.272	
	Unknown	11							
Type of local treatment of all bone metastases									
	Surgery only	9	0.778	0.139	0.519	0.176	0.130	0.121	0.913
	Radiotherapy involved	3	0.333	0.272	0.333	0.272	0.333	0.272	
	Not all bone metastases treated locally/unknown	47							
Macroscopically complete resection of all bone metastases									
	No	57	0.424	0.065	0.129	0.046	0.064	0.035	0.062
	Yes	9	0.778	0.139	0.519	0.176	0.130	0.121	
	Unknown	3							
Radiotherapy as part of local treatment of all unresected bone metastases									
	No	55	0.436	0.067	0.163	0.052	0.041	0.028	0.250
	Yes, radiotherapy involved	3	0.333	0.272	0.333	0.272	0.333	0.272	
	Unknown	11							

* *p*-value, log-rank test.

## Data Availability

The authors confirm that the data supporting the findings of this study are available within the article.

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
