# Peer review of "Primary Multi-Systemic Metastases in Osteosarcoma: Presentation, Treatment, and Survival of 83 Patients of the Cooperative Osteosarcoma Study Group"

_cancers, 2024, doi:10.3390/cancers16020275_

Round 1
Reviewer 1 Report
Comments and Suggestions for Authors
A well written paper that describes with clarity the ominous outcome of particularly aggressive osteosarcomas. The information provided are not new, however, and no significant clues are suggested on the pathophysiology of the disease in this subset of patients or how to successfully overcome the progression of this cancer.
Author Response
Thank you very much for taking the time to review this manuscript. As you very accurately describe, this retrospective analysis unfortunately found no evidence of a truly promising therapeutic approach in the patient cohort analysed. It is precisely for this reason that we ultimately conclude that new therapeutic options are urgently needed. Nevertheless, we believe that the results of this study are relevant for clinical practitioners: Although the prognosis for those affected is inevitably poor, it is not completely hopeless, even with initially advanced disease - especially if a complete remission appears technically possible (be it through surgical resection of all metastases or in combination with radiotherapy of bone metastases).
Reviewer 2 Report
Comments and Suggestions for Authors
Thank you for inviting me to evaluate the article, “Primary multi-systemic metastases in osteosarcoma: presentation, treatment, and survival of 83 patients of the Cooperative Osteosarcoma Study Group”. This study aimed to identify prognostic factors and to evaluate the effect of various therapeutic interventions in osteosarcoma patients with primary metastases.
Although the progress of neoadjuvant/adjuvant chemotherapies along with limb-sparing surgery has improved both the prognosis of patients with localized osteosarcoma, however the prognosis of the patients with metastatic or recurrent or chemotherapy-resistant lesions has remained very low.
Generally, the work is very interesting and valuable in the field of cancer research, and is of interest to the broad readership of Cancers. This article should be accepted for publication in Cancers. There are some questions, that should be addressed.
Comments:
1. The authors discussed that complete resection of the bone metastases resulted in a more favorable prognosis. In patients with multiple metastases, complete resection of the metastatic lesions in any sites should be often difficult. Does “complete resection” mean a complete resection and reconstruction using prosthesis/bone grafting ? Or, does it include a resection as much plus bone grafting/radiation etc.?
2. Is this study, an elevated AP value at initial diagnosis is associated with poorer prognosis. So, in patients with a better prognosis, was the AP value decreased by therapeutic interventions ?
Author Response
Thank you very much for taking the time to review this manuscript. Please find the detailed responses below and the corresponding revisions highlighted (green) in the re-submitted files.
Comments to the Author
1. The authors discussed that complete resection of the bone metastases resulted in a more favorable prognosis. In patients with multiple metastases, complete resection of the metastatic lesions in any sites should be often difficult. Does “complete resection” mean a complete resection and reconstruction using prosthesis/bone grafting ? Or, does it include a resection as much plus bone grafting/radiation etc.?
Response: We could (only) demonstrate a trend for better survival for patients undergoing a complete resection of all bone metastases - possibly due to the rather limited cohort size. A complete (macroscopic) resection refers to a resection that is performed within the scope of a surgical resection. We have adapted the wording (lines xxx and xxx) to make this clearer. On the other hand, a complete local treatment of all bone metastases refers to any local treatment including surgical resection and/or radiotherapy of bone metastases.
2. In this study, an elevated AP value at initial diagnosis is associated with poorer prognosis. So, in patients with a better prognosis, was the AP value decreased by therapeutic interventions?
This is an interesting question we would very much like to answer. But unfortunately, here our study is limited by its retrospective nature: While the AP values prior to the start of therapy were systematically recorded over many years, we only have very fragmentary data on AP values at a defined later timepoint of therapy. Unfortunately, we therefore cannot answer this question.
Reviewer 3 Report
Comments and Suggestions for Authors
In the present study, the authors reported a study based on a group of 83 patients with osteosarcoma, trying to identify potential prognostic factors and to evaluate the impacts of therapeutic interventions. All the information provided is detailed, but the finding of this study is limited in novelty. Besides, there are other major limitations in the manuscript listed as followed.
1. The results of the manuscript are not well organized and make it hard to read. First, the tables and figures are not cited in the manuscript. Second, All the tables and figures are not arranged according to the order that appeared in the results. Third, each part of results lack the summary and links between different parts.
2. The conclusions in this study are limited in novelty. The impacts of Surgical resection, radio-therapy, and standard chemotherapy are well-known to us, what is the most novel finding through this study, the author may need to think more about it as well as the meanings from this study.
3. Further discussion is needed rather than re-description of the results.
4. Most references are published before 2020, are there any recent references related to the clinical situation about osteosarcoma? The authors had better add more recent references.
Author Response
Thank you very much for taking the time to review this manuscript. Please find the detailed responses below and the corresponding revisions highlighted in the re-submitted files.
Comments to the Author
1. The results of the manuscript are not well organized and make it hard to read. First, the tables and figures are not cited in the manuscript. Second, All the tables and figures are not arranged according to the order that appeared in the results. Third, each part of results lack the summary and links between different parts.
Response: Thank you for pointing this out. In agreement with the reviewers remark, we edited the manuscript accordingly: We cited both the tables and the figures, arranged them according to the order they appear in the results and have adapted the text with regard to summaries and links between different parts. Please find all the changes made in this regard highlighted in the "results" section of the revised manuscript (lines 166f, 171, 203, 212f, 236ff, 241, 242ff, 248f, 251f).
2. The conclusions in this study are limited in novelty. The impacts of surgical resection, radio-therapy, and standard chemotherapy are well-known to us, what is the most novel finding through this study, the author may need to think more about it as well as the meanings from this study.
Response: As you very accurately describe, this retrospective analysis unfortunately found no evidence of a new and truly promising therapeutic approach in the patient cohort analysed. It is precisely for this reason that we ultimately conclude that new therapeutic options are urgently needed. Nevertheless, we believe that the results of this study are relevant for clinical practitioners: Although the prognosis for those affected is inevitably poor, it is not completely hopeless, even with initially advanced disease - especially if a complete remission appears technically possible (be it through surgical resection of all metastases or in combination with radiotherapy of bone metastases).
3. Further discussion is needed rather than re-description of the results.
Response: We have, accordingly, revised the discussion. Please find the multiple changes made highlighted in the "discussion" section of the revised manuscript (line 316ff, 338ff, 345ff, 348ff, 386ff, 392ff).
4. Most references are published before 2020, are there any recent references related to the clinical situation about osteosarcoma? The authors had better add more recent references.
Response: Thank you for pointing that out - we added more recent references on the clinical situation of osteosarcoma throughout our manuscript. Please find all the references added highlighted (green for references published since 2020) in the "reference" section of the revised manuscript (reference number 10-12, 23, 27, 47-51).
Round 2
Reviewer 3 Report
Comments and Suggestions for Authors
The authors have revised according to the comments and the current vision is fine for publication. Thank you!